# Risk of Dementia in Patients Who Underwent Surgery under Neuraxial Anesthesia: A Nationwide Cohort Study

**DOI:** 10.3390/jpm11121386

**Published:** 2021-12-20

**Authors:** Young-Suk Kwon, Jae-Jun Lee, Sang-Hwa Lee, Chulho Kim, Hyunjae Yu, Jong-Hee Sohn, Dong-Kyu Kim

**Affiliations:** 1Department of Anesthesiology and Pain Medicine, College of Medicine, Chuncheon Sacred Heart Hospital, Hallym University College of Medicine, Chuncheon 24253, Korea; gettys@hallym.or.kr (Y.-S.K.); iloveu59@hallym.or.kr (J.-J.L.); 2Institute of New Frontier Research, Division of Big Data and Artificial Intelligence, Chuncheon Sacred Heart Hospital, Hallym University College of Medicine, Chuncheon 24253, Korea; neurolsh@hallym.or.kr (S.-H.L.); gumdol52@hallym.or.kr (C.K.); yunow@hallym.or.kr (H.Y.); 3Department of Neurology, Chuncheon Sacred Heart Hospital, Hallym University College of Medicine, Chuncheon 24253, Korea; 4Department of Otorhinolaryngology-Head and Neck Surgery, Chuncheon Sacred Heart Hospital, Hallym University College of Medicine, Chuncheon 24253, Korea

**Keywords:** neuraxial anesthesia, dementia, Alzheimer’s disease, vascular dementia, nationwide cohort study

## Abstract

The incidence of dementia in patients with surgery under neuraxial anesthesia and the possibility of surgery under neuraxial anesthesia as a risk factor for dementia were investigated. We performed a retrospective matched cohort study with nationwide, representative cohort sample data of the Korean National Health Insurance Service in South Korea between 1 January 2003, and 31 December 2004. The participants were divided into control (*n* = 4488) and neuraxial groups (*n* = 1122) using propensity score matching. After 9 years of follow-up, the corresponding incidences of dementia were 11.5 and 14.8 cases per 1000 person-years. The risk of dementia in the surgery under neuraxial group was 1.44-fold higher (95% confidence interval [95%CI], 1.17–1.76) than that in the control group. In the subgroup analysis of dementia, the risk of Alzheimer’s disease in those who underwent surgery under neuraxial anesthesia was 1.48-fold higher (95%CI, 1.17–1.87) than that in those who did not undergo surgery under anesthesia. Our findings suggest that patients who underwent surgery under neuraxial anesthesia had a higher risk of dementia and Alzheimer’s disease than those who did not undergo surgery under neuraxial anesthesia.

## 1. Introduction

Neuraxial anesthesia techniques include spinal, epidural, and combined spinal-epidural anesthesia. The anesthesia level needed in a specific surgery is determined by the dermatome level of the skin incision part and the level needed for surgical manipulation, but neuraxial anesthesia is commonly used in surgery of the lower abdomen and lower extremities and provides an alternative to general anesthesia when appropriate [1]. Previous observational and randomized studies have suggested that surgery under neuraxial anesthesia is associated with better results than surgery under general anesthesia for some procedures, although it lacked high-quality evidence [2] and did not have more significant advantages over other specific types of anesthesia [3,4]. In addition, research findings on the association between anesthesia and dementia have been inconsistent [5,6,7,8]. Postoperative cognitive decline is a general anesthesia complication [9], which may develop in up to 80% and 26% of patients who undergo cardiac surgeries and non-cardiac surgeries, respectively [10]. There is a perception that postoperative cognitive decline may increase the risk of dementia and Alzheimer’s disease [9]; thus, some patients prefer surgery under neuraxial anesthesia to general anesthesia. However, it is difficult to separate and evaluate the effects of surgery and anesthesia on cognitive function. Neuraxial anesthesia may expose patients to a certain degree of postoperative cognitive decline and dementia. In the present study, we aimed to evaluate the incidence of dementia in patients who underwent surgery under neuraxial anesthesia and investigate the potential of surgery under neuraxial anesthesia as a risk factor for dementia using nationwide, representative cohort sample data.

## 2. Materials and Methods

### 2.1. National Sample Cohort

The Korean National Health Insurance Service (KNHIS) has provided mandatory health coverage to the South Korean population since 1989. A unique identification number assigned to each South Korean resident at birth prevents the omission or duplication of healthcare data. With the integration of Medical Aid data into the KNHIS database in 2006, this database comprises the entire population of South Korea. Therefore, usage of the KNHIS database eliminates selection bias. The KNHIS database contains nearly all medical data, including the diagnostic codes according to the Korean Classification of Diseases, which are similar to those of the International Classification of Diseases. The present study utilized a representative sample of 1,025,340 adults from the 2002–2013 KNHIS–National Sample Cohort (NSC) in South Korea (NHIS-2018-2-258). This dataset accounts for approximately 2.2% of the South Korean population in 2002. Stratified random sampling was performed using 1476 strata by age (18 groups), sex (two groups), and income level (41 groups: 40 health insurance groups and one medical aid beneficiary) of the South Korean population of 46 million in 2002. Additionally, the KNHIS–NSC contains data of all health services, including hospital visits (inpatient and outpatient), medical procedures, drug prescriptions, hospital diagnoses, and demographic information (including sex, age, household income, and mortality) during the study period of 2002–2013. To date, numerous previous studies have been published using these data.

### 2.2. Study Setting and Participants

This study was approved by the Institutional Review Board of Hallym Medical University Chuncheon Sacred Hospital (IRB No. 2021-08-006), and the need for written informed consent was waived because the KNHIS-NSC dataset comprised de-identified secondary data for research purposes. The study cohort comprised patients who underwent surgery under neuraxial anesthesia during the index period (1 January 2003, to 31 December 2004) and were aged over 55 years at enrollment. To remove any potential pre-existing cases of anesthesia, we established a washout period (1 January 2002, to 31 December 2002). Additionally, we excluded the following patients: (1) underwent surgery under neuraxial anesthesia before and after the index period; (2) underwent surgery under other anesthesia besides neuraxial anesthesia from 2002 to 2013; (3) with a history of brain and heart surgery from 2002 to 2013; (4) diagnosed with dementia before and during the index period; and (5) died during the index period. The comparison group (patients who did not receive anesthesia) comprised randomly selected propensity score-matched individuals from the remaining cohort registered in the database (four for each patient who underwent surgery under neuraxial anesthesia) between 2003 and 2004. The study and comparison groups were matched by variables such as sociodemographic factors (age, sex, residential area, and household income), Charlson comorbidity index, and the enrollment date. The schematic description of the cohort design is presented in Figure 1.

### 2.3. Predictor and Outcome Variables

We collected data for patients diagnosed with dementia (Alzheimer’s disease [F00, G30], vascular dementia [F01], and others [F02, F03]) from 2002 to 2013. Detailed patient characteristics, including sex, age, residence, household income, and Charlson comorbidity index, are presented in Table 1. The study population was divided into three age groups (55–64, 65–74, and ≥75 years), three income groups (low: ≤30%, middle: 30.1–69.9%, and high: ≥70% of the median), three residential areas (first area: Seoul, the largest metropolitan region in South Korea; second area: other metropolitan cities in South Korea; and third area: small cities and rural areas), and three Charlson comorbidity index group (0, 1, and ≥2).The Charlson comorbidity index, developed based on medical record data, converted 19 diseases into ICD-10 codes for application to administrative data, and the list of Charlson comorbidity index and weight are summarized in Appendix A
Table A1. The study endpoint was the death of a participant or the incidence of dementia. The characteristics of all patients who had no events and who were alive until 31 December 2013 are shown in Appendix A
Table A2. The risks of dementia in the surgery under neuraxial anesthesia and comparison groups were compared using person-years at risk, which was defined as the duration between the date of enrollment and the patient’s respective endpoint.

### 2.4. Statistical Analysis 

We employed one-to-four propensity score-matching according to age, sex, residential area, household income, and comorbidities. The incidence rates per 1000 person-years for dementia were obtained by dividing the number of patients with dementia by person-years at risk. The overall disease-free survival rate was determined using Kaplan–Meier survival curves for the entire observation period. To evaluate the risk association between surgery under neuraxial anesthesia and dementia, we used Cox proportional hazard regression to calculate the hazard ratio and 95% confidence intervals (CI), adjusting for other predictor variables. As a subgroup analysis, we evaluated hazard ratios of dementia according to sex, age, Charlson comorbidity index, type of surgery, and dementia type among the sample patient. All statistical analyses were performed using R version 3.3.1 (R Foundation for Statistical Computing, Vienna, Austria) with a significance level of 0.05. 

**Table 1 jpm-11-01386-t001:** Characteristics of the study subjects.

Variables		Comparison (*n* = 4488)	Surgery under Neuraxial Anesthesia (*n* = 1122)	*p* Value
Sex	Male	2044 (45.5%)	511 (45.5%)	1.000
	Female	2444 (54.5%)	611 (54.5%)	
Ages	55–64	1892 (42.2%)	473 (42.2%)	1.000
	65–74	1596 (35.6%)	399 (35.6%)	
	≥75	1000 (22.3%)	250 (22.3%)	
Residence	Seoul	908 (20.2%)	227 (20.2%)	1.000
	Second area	940 (20.9%)	235 (20.9%)	
	Third area	2640 (58.8%)	660 (58.8%)	
Household	Low (0–30%)	992 (22.1%)	248 (22.1%)	1.000
	Middle (30–70%)	1424 (31.7%)	356 (31.7%)	
	High (70–100%)	2072 (46.2%)	518 (46.2%)	
CCI	0	2472 (55.1%)	618 (55.1%)	1.000
	1	1024 (22.8%)	256 (22.8%)	
	≥2	992 (22.1%)	248 (22.1%)	

Comparison, subjects without anesthesia; Seoul, the largest metropolitan area; second area, other metropolitan cities; third area, other areas; CCI, Charlson comorbidity index.

## 3. Results

### 3.1. Effects of the Surgery under Neuraxial Anesthesia on the Incidence of Dementia among Patients Aged over 55 Years 

The present study comprised 1122 patients who underwent surgery under neuraxial anesthesia and 4488 comparison participants (patients who did not undergo surgery under anesthesia). The two cohorts (the surgery under neuraxial anesthesia group and the comparison group) had similar distributions of sex, age, residential area, household income, and Charlson comorbidity index, meaning that each variable was matched appropriately between the two groups (Table 1, Figure 2). The overall incidence of dementia was higher in the surgery under neuraxial anesthesia group (14.8 per 1000 person-years) than in the comparison group (11.5 per 1000 person-years) (Table 2).

### 3.2. Hazard Ratios of Dementia in Patients Aged over 55 Years and Who Underwent Surgery under Neuraxial Anesthesia

Figure 3 presents the Kaplan–Meier survival curves with log-rank tests for the cumulative hazard plot of specific disease-free between comparison and surgery under neuraxial anesthesia groups. The results of the log-rank test indicated that patients who underwent surgery under neuraxial anesthesia developed dementia more frequently than those who did not undergo surgery under anesthesia during the 9-year follow-up period. In the subgroups analysis, patients who underwent surgery under neuraxial anesthesia developed Alzheimer’s disease more frequently than those who did not undergo surgery under anesthesia during the 9-year follow-up period (Figure 4).

**Figure 2 jpm-11-01386-f002:**
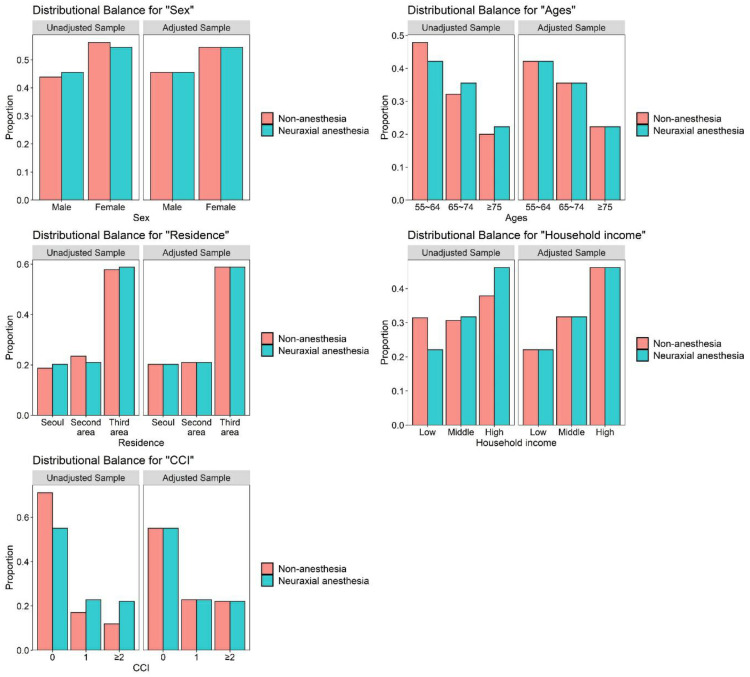
Balance plot for five variables before and after matching.

**Table 2 jpm-11-01386-t002:** Incidence per 1000 person-years and HR (95% CIs) of dementia during the follow-up period.

Variables	N	Case	Incidence	Unadjusted HR (95% CI)	Adjusted HR (95% CI)	*p* Value
Group						
Comparison	4488	442	11.49	1.00 (ref)	1.00 (ref)	
Surgery under neuraxial anesthesia	1122	121	14.78	1.40 (1.14–1.72) **	1.44 (1.17–1.76) ***	<0.001
Sex						
Male	2555	192	9.33	1.00 (ref)	1.00 (ref)	
Female	3055	371	14.23	1.51 (1.27–1.80) ***	1.32 (1.10–1.57) **	0.002
Ages (years)						
55–64	2365	77	3.40	1.00 (ref)	1.00 (ref)	
65–74	1995	240	14.35	4.44 (3.43–5.74) ***	4.26 (3.29–5.51) ***	<0.001
≥75	1250	246	33.66	11.90 (9.20–15.40) ***	11.36 (8.77–14.72) ***	<0.001
Residence						
Seoul	1135	99	9.99	1.00 (ref)	1.00 (ref)	
Second area	1175	126	12.92	1.31 (1.01–1.70) *	1.19 (0.92–1.56)	0.188
Third area	3300	338	12.52	1.27 (1.02–1.59) *	1.0aq7 (0.85–1.34)	0.573
Household income						
Low (0–30%)	1240	132	13.12	1.00 (ref)	1.00 (ref)	
Middle (30–70%)	1780	151	10.09	0.76 (0.60–0.96) *	0.87 (0.69–1.10)	0.239
High (70–100%)	2590	280	12.94	0.98 (0.79–1.20)	0.99 (0.80–1.21)	0.894
CCI						
0	3090	283	10.54	1.00 (ref)	1.00 (ref)	
1	1280	163	15.24	1.47 (1.21–1.78) ***	1.37 (1.13–1.66) **	0.001
≥2	1240	117	12.82	1.25 (1.01–1.55) *	1.21 (0.97–1.50)	0.091

Seoul, the largest metropolitan area; second area, other metropolitan cities; third area, other areas; CCI, Charlson comorbidity index; HR, hazard ratio; CI, confidence interval. * *p* < 0.05, ** *p* < 0.010, and *** *p* < 0.001.

**Figure 3 jpm-11-01386-f003:**
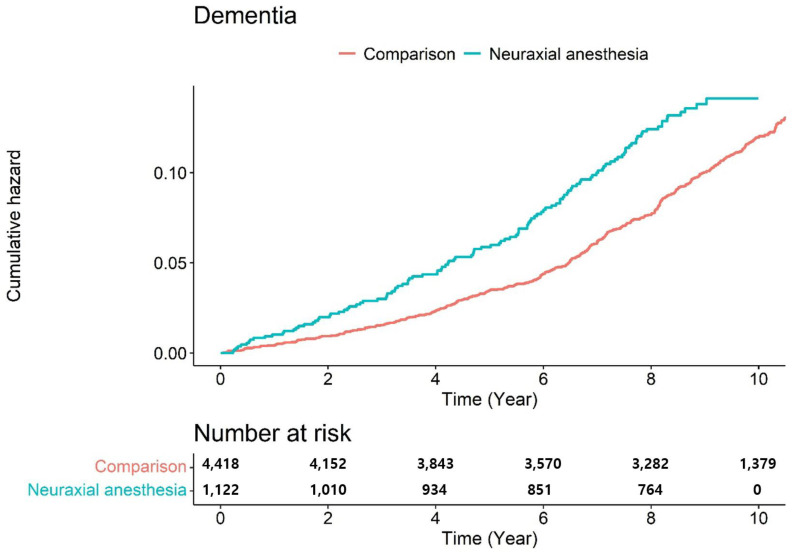
Risk of development of dementia disease between neuraxial anesthesia and comparison (non-anesthesia).

**Figure 4 jpm-11-01386-f004:**
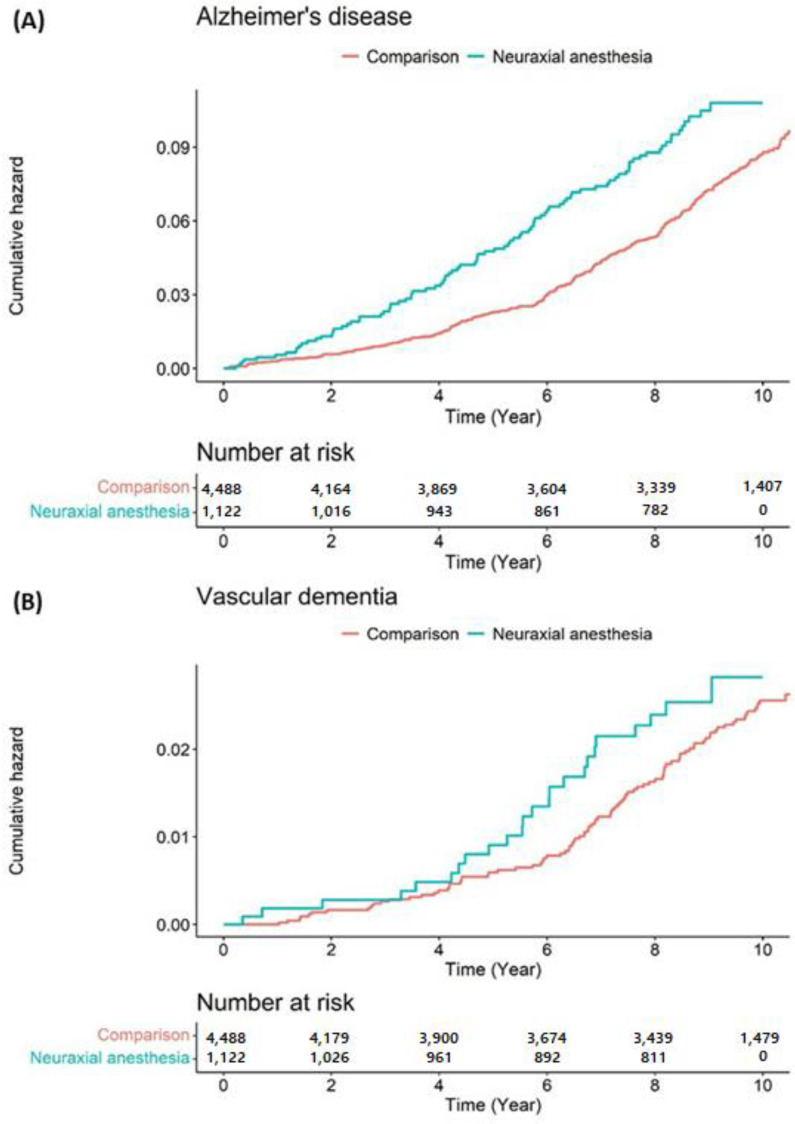
Cumulative hazard plot of specific between neuraxial anesthesia patient group and comparison group: (**A**) Alzheimer’s disease. (**B**) Vascular disease.

We used simple and multiple Cox regression models to analyze the hazard ratio for the development of dementia (Table 2). After adjusting for sociodemographic factors and Charlson comorbidity index, we found that surgery under neuraxial anesthesia was associated with prospective dementia development with an adjusted hazard ratio of 1.44 (95% CI, 1.17–1.76). In addition, female and increasing age were significantly associated with the prospective development of dementia. 

In the subgroup analysis, we observed no significant difference in the hazard ratios of dementia by sex between surgery under neuraxial anesthesia and non-anesthesia groups. Additionally, the adjusted hazard ratio for dementia development among the older (≥75) patients who underwent surgery under neuraxial anesthesia during the 9-year follow-up period was 1.60 (95% CI: 1.19–2.15). The adjusted hazard ratio for dementia development among patients who underwent surgery under neuraxial anesthesia with Charlson comorbidity index of 0 during the 9-year follow-up period was 1.62 (95% CI: 1.23–2.15) (Table 3). Moreover, the adjusted hazard ratio of developing Alzheimer’s disease in patients aged ≥55 who underwent surgery under neuraxial anesthesia during the 9-year follow-up period was 1.52 (95% CI, 1.20–1.92) compared with the patients who did not undergo surgery under anesthesia; however, we did not find any association between surgery under neuraxial anesthesia and vascular dementia (adjusted hazard ratio of 1.29; 95% CI, 0.82–2.03) in patients ≥55 who underwent surgery under neuraxial anesthesia (Table 4). Table 5 summarizes the incidence per 1000 person-years and risk of dementia, Alzheimer’s disease, and vascular dementia according to type of surgery.

## 4. Discussion

The present study investigated the incidence of dementia in older adults who underwent surgery under neuraxial anesthesia and compared the incidence risk of dementia between those who underwent surgery under neuraxial anesthesia and those who did not undergo surgery under neuraxial anesthesia in Korea. The corresponding incidences of dementia during the 9-year follow-up period were 14.8 and 11.5 cases per 1000 person-years. The incidence risk of dementia in older adults who underwent surgery under neuraxial anesthesia was 1.44-fold higher than that in older adults who did not undergo surgery under neuraxial anesthesia, even after adjusting for several risk factors, including age, sex, residence, household income, and comorbidities. Similarly, the incidence risk of Alzheimer’s disease in older adults who underwent surgery under neuraxial anesthesia was 1.52-fold higher than that in older adults who did not undergo surgery under neuraxial anesthesia. Surgery under neuraxial anesthesia increased the risk of dementia in individuals with a Charlson comorbidity index of 0, which suggests that surgery under neuraxial anesthesia can increase the risk of dementia in the absence of comorbidities. Patients who underwent surgery under neuraxial anesthesia had elevated risk of dementia and Alzheimer’s disease regardless of the type of surgery, which suggests that neuraxial anesthesia increases the risk of dementia. 

Several studies have reported an association between anesthesia and dementia, but the effect of general anesthesia on dementia remains controversial. A systematic review and meta-analysis of 15 case-control studies did not find a significant association between anesthesia and Alzheimer’s disease [11]. However, studies investigating the association between dementia and surgery under neuraxial anesthesia are limited. There are some studies on the postoperative cognitive dysfunction or delirium after surgery under neuraxial anesthesia. Ehsani et al. reported that the incidence of postoperative delirium and early cognitive disorder was higher in general anesthesia than in spinal anesthesia [12]. However, Zhang et al. showed that spinal anesthesia + isoflurane was associated with a higher incidence of dementia than spinal anesthesia, but there was no difference between spinal anesthesia and spinal anesthesia + desflurane [13]. Silbert et al. reported no significant difference in the rate of postoperative cognitive dysfunction between general anesthesia and spinal anesthesia [14]. Aiello et al. reported that exposure to general anesthesia or neuraxial anesthesia was not associated with dementia or Alzheimer’s disease in community-dwelling members of the Adult Changes in Thought cohort aged 65 years and above and free of dementia at baseline [9]. However, their study had some limitations. First, data were collected through interviews and patient recall. Second, the demographic composition was primarily white, middle class, well-educated patients, which did not reflect the general population. In contrast, we investigated the general population with long-term follow-up, and our results showing high incidence in older age, females, and low-income earners and those residing in rural areas are similar to previous studies [15,16].

It is unclear why older adults with surgery under neuraxial anesthesia have a higher incidence of dementia than those without surgery under neuraxial anesthesia. Our results showed that the effect on dementia is more associated with Alzheimer’s disease than vascular dementia. The direct effect of surgery under neuraxial anesthesia on the pathology of Alzheimer’s disease is difficult to explain, but the neurotoxicity of local anesthetics can be considered. In the spinal canal after spinal anesthesia, lidocaine-induced apoptosis, and higher concentrations of lidocaine induced necrosis and non-specific apoptosis [17,18]. However, the effects of spinal anesthesia on the brain have not been studied, and the effects of local anesthesia may be limited to the specific organ or tissue into which the anesthetic is injected. Fathy et al. reported that lidocaine and bupivacaine can lead to postoperative cognitive impairment after cataract surgery under local anesthesia, but they did not convincingly demonstrate whether this outcome was a definitive effect of local anesthetics [19].

Postoperative events can be a risk factor for dementia. In animals, postoperative cognitive dysfunction is associated with postoperative cytokine-induced inflammation in the hippocampus [20,21]. Various proinflammatory cytokines, such as tumor necrosis factor alpha, maintain a state of chronic neuroinflammation, resulting in postoperative cognitive impairment and postoperative delirium [22,23]. Surgery may induce astrogliosis, β-amyloid accumulation, and τ phosphorylation in the elderly, which may be associated with the cognitive decline seen in postoperative cognitive dysfunction [24]. The decreased functional connectivity of the executive control network and its anticorrelation with the default mode network may contribute to executive function deficits following surgery [25]. The brain drives the surgical stress response by initiating changes in neuroendocrine balance, but changes in homeostasis can lead to postoperative cognitive impairment (POCD) [26]. Changes in the metabolism before and after surgery can be related to postoperative delirium [27]. Perioperative management may be associated with a risk of dementia. Preoperative fasting, body temperature control, and blood pressure management are associated with the risk of postoperative cognitive disorder [28]. Patients with a high risk of postoperative neurological complications require frailty screening with preoperative cognitive screening for the best perioperative neurological results [29].

Reduction of cerebrospinal fluid pressure due to cerebrospinal fluid loss in the epidural space after spinal anesthesia is not uncommon [30]. In older people, there may be a persistent and significant decrease in cerebrospinal fluid pressure [31]. Patients with low cerebrospinal fluid flow have significantly reduced memory, visual construction, and verbal fluency. Alterations in cerebrospinal fluid flow may contribute to some of the cognitive deficits observed in Alzheimer’s disease patients [32]. Injection of the drug into the lumbar epidural space compresses the dural sac, changes the compliance of the spinal subarachnoid space, and moves cerebrospinal fluid upwards towards the skull. Depending on the amount of the drug, intracranial pressure may increase [33]. The amount can be increased with prolonged anesthesia or with a patient-controlled analgesia catheter administered after nerve axis anesthesia. Increased intracranial pressure may be associated with cognitive impairment [34]. Spinal anesthesia triggers cerebral vasodilation [35] and may raise intracranial pressure [36]. In addition, the ability to control cerebral blood flow decreases in the elderly [37].

The elevated risk of cognitive deficits may be explained by the effect of neuraxial anesthesia on the risk factors for Alzheimer’s disease. Cerebrovascular disease is associated with decreasing cognitive function, and it lowers the threshold of clinical dementia in patients with neuropathological diagnosis of Alzheimer’s disease [38,39,40]. Many older patients take anticoagulants for atrial fibrillation, coronary artery disease, and thromboembolism cases. The anticoagulants should be discontinued before surgery under neuraxial anesthesia. However, perioperative discontinuation of anticoagulants for several days can cause acute ischemia, which raises the risk of acute ischemic stroke [41,42]. If only the small blood vessels are affected, it may be difficult to deny that cerebral ischemia has occurred, even if there are no symptoms of cerebral ischemia after surgery under anesthesia [43]. The effect of surgery on dementia should also be considered. Neuraxial anesthesia slows the stress response during surgery, reduces intraoperative bleeding, and reduces the frequency of postoperative thromboembolism. In addition, considering the advantages of reducing morbidity and mortality in high-risk patients and controlling post-operative pain, surgery may have a significant impact on dementia [1]. After surgery, patients have limited physical activity, which may increase the risk of dementia compared to active physical activity [44]. In particular, many surgeries of the lower extremities are performed under neuraxial anesthesia, and the patients may have prolonged immobility depending on the postoperative condition

Approximately 50 million people worldwide have dementia, and an estimated 5–8% of the general population over the age of 60 years develops dementia [45]. The treatment for dementia is expensive, and the earlier the onset, the higher the cost of care. Considering that most types of dementia, such as Alzheimer’s disease, have slow progression, the associated risk factors should be investigated and controlled [46]. Therefore, the risk of dementia should be considered when selecting the anesthesia type and surgery. Although surgery under neuraxial anesthesia may be associated with dementia risk factors, surgery under neuraxial anesthesia is thought to have little relevance to dementia because patients are conscious during anesthesia. However, despite the increase in surgical demands at old age [47], and considering that studies on dementia require long-term follow-up, studies on the effect of neuraxial anesthesia on dementia have been limited.

The present study has some limitations. First, we could not determine the anesthesia and surgery experience of patients before the index period because we categorized patients according to the presence of neuraxial anesthesia experience during the index period. Considering that dementia progresses slowly, anesthesia and surgery experience before the index period might have affected the development of dementia. However, to overcome this challenge, we only enrolled subjects who were 55 years of age or older and investigated the effect of surgery under neuraxial anesthesia in old age. Additionally, this study had a one-year washout period for surgery under anesthesia and excluded patients with additional surgery under anesthesia after the index period to ensure that only the effect of surgery under neuraxial anesthesia during the index period was evaluated. Moreover, we perfectly matched the surgery under neuraxial anesthesia and non-surgery under neuraxial anesthesia groups using propensity scores with several variables, including age, sex, residence, household income, and comorbidities, and the effect of these variables on dementia was similar to that of previous studies. Second, it was difficult to evaluate the effect of various procedures, including surgeries and local anesthesia, on our findings. We evaluated the effects of anesthesia and surgery on dementia as a series of procedures. However, surgery may affect immediate survival and long-term outcomes [48]. There are many types of surgery, and the corresponding intra- and post-operative effects may be different. The comparison group was non-anesthesia regardless of surgery. We classified the two groups by code of anesthesia, but there was no separate code for local anesthesia in KNHIS-NSC. The comparison group may include simple surgery under local anesthesia. Local anesthesia is usually administered for simple procedures, including skin surgery, open wound repair, abscess drainage, foreign body removal from the skin, vascular access, and dental procedures [49,50,51]. The direct effect on the brain by local anesthesia for simple surgery may be limited. In addition, intensive postoperative care is not needed because complications of simple surgery are rare, and active pain control is usually not required because the surgical wound is small. Although we used the KNHIS–NSC dataset in the present study, it has been established for medical service claims and reimbursement, not for research. This dataset lacked sufficient data on procedures, including surgeries and local anesthesia. Additional studies are needed to evaluate the effects of each surgery and local anesthesia. Third, our results are likely to be further biased due to confounding by indication or postoperative events, that is, patients who received neuraxial anesthesia may have done so because of unmeasured differences between groups, such as frailty, patient or clinician preference, or traumatic brain injury and stroke during the follow-up period. We matched the Charlson comorbidity index between the two groups to reduce the confounding as much as possible. Fourth, we could not validate the diagnosis of dementia, Alzheimer’s disease, and vascular dementia with the ICD code. Thus, we may have underestimated cognitive dysfunction or dementia diagnosis. Further studies are needed to analyze hospital-based or registry-based datasets that include clinical cognitive function tests to confirm our findings.

## 5. Conclusions

The present study showed that older patients who underwent surgery under neuraxial anesthesia had a higher risk of developing dementia. Especially patients who underwent surgery under neuraxial anesthesia showed a higher risk of Alzheimer’s disease. Anesthesia is accompanied by a series of processes leading to surgery and post-operative care, and since we evaluated the effect of surgery under neuraxial anesthesia on dementia, it is still insufficient to determine the effects of surgery and neuraxial anesthesia on dementia, respectively. However, our findings reveal new insights that surgery under neuraxial anesthesia can be considered a possible risk factor for dementia.

## Figures and Tables

**Figure 1 jpm-11-01386-f001:**
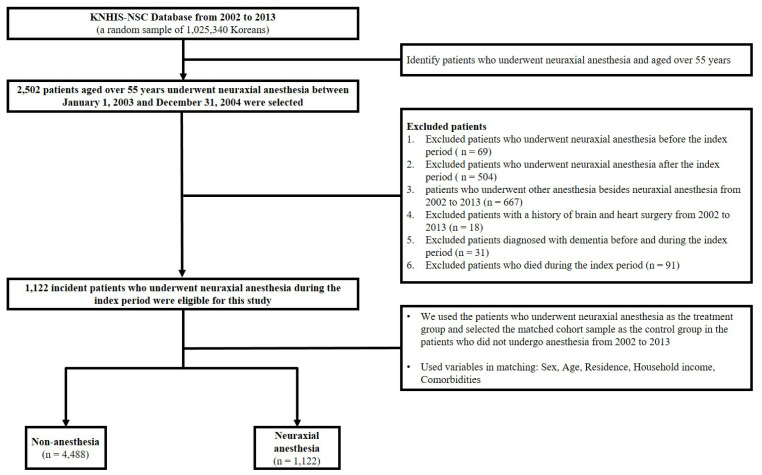
Schematic description of study design.

**Table 3 jpm-11-01386-t003:** Hazard ratios of dementia by sex, age, and comorbidity score between surgery under neuraxial anesthesia and comparison (non-anesthesia).

Sex	Male				Female			
	Comparison		Surgery under Neuraxial Anesthesia		Comparison		Surgery under Neuraxial Anesthesia	
Unadjusted HR (95% CI)	1.00 (ref)		1.54 (1.09–2.17) *		1.00 (ref)		1.33 (1.03–1.71) *	
Adjusted HR (95% CI)	1.00 (ref)		1.60 (1.13–2.25) **		1.00 (ref)		1.36 (1.05–1.75) *	
Ages	55–64			65–74		≥75		
	Comparison	Surgery under neuraxial anesthesia		Comparison	Surgery under neuraxial anesthesia	Comparison		Surgery under neuraxial anesthesia
Unadjusted HR (95% CI)	1.00 (ref)	1.32 (0.74–2.35)		1.00 (ref)	1.30 (0.94–1.80)	1.00 (ref)		1.60 (1.19–2.16) **
Adjusted HR (95% CI)	1.00 (ref)	1.33 (0.75–2.37)		1.00 (ref)	1.31 (0.95–1.81)	1.00 (ref)		1.60 (1.19–2.15) **
CCI	0			1		≥2		
	Comparison	Surgery under neuraxial anesthesia		Comparison	Surgery under neuraxial anesthesia	Comparison		Surgery under neuraxial anesthesia
Unadjusted HR (95% CI)	1.00 (ref)	1.55 (1.17–2.05) **		1.00 (ref)	1.33 (0.90–1.96)	1.00 (ref)		1.18 (0.73–1.88)
Adjusted HR (95% CI)	1.00 (ref)	1.62 (1.23–2.15) ***		1.00 (ref)	1.39 (0.94–2.05)	1.00 (ref)		1.11 (0.69–1.77)

CCI, Charlson comorbidity index; HR, hazard ratio; CI, confidence interval. * *p* < 0.05, ** *p* < 0.010, and *** *p* < 0.001.

**Table 4 jpm-11-01386-t004:** Incidence per 1000 person-years and HR (95% CI) of specific diseases (Alzheimer’s disease and vascular dementia).

Variables	N	Case	Incidence	Unadjusted HR (95% CI)	Adjusted HR (95% CI)	*p* Value
Alzheimer’s Disease						
Comparison	4488	328	8.45	1.00 (ref)	1.00 (ref)	
Surgery under neuraxial anesthesia	1122	92	11.13	1.48 (1.17–1.87) **	1.52 (1.20–1.92) ***	<0.001
Vascular Dementia						
Comparison	4488	95	2.40	1.00 (ref)	1.00 (ref)	
Surgery under neuraxial anesthesia	1122	24	2.84	1.27 (0.81–2.01)	1.29 (0.82–2.03)	0.273

HR, hazard ratio; CI, confidence interval. ** *p* < 0.010, and *** *p* < 0.001.

**Table 5 jpm-11-01386-t005:** Incidence per 1000 person-years and HR (95% CI) of dementia, Alzheimer’s disease and vascular dementia according to surgery type.

Variables	N	Case	Incidence	Unadjusted HR (95% CI)	Adjusted HR (95% CI)	*p* Value
Dementia						
Comparison	4488	442	11.49	1.00 (ref)	1.00 (ref)	
Minor	1032	107	13.95	1.32 (1.06–1.63) *	1.40 (1.13–1.74) **	0.002
Major	90	14	27.06	2.76 (1.62–4.71) ***	1.78 (1.04–3.04) *	0.036
Alzheimer’s disease						
Comparison	4488	328	8.45	1.00 (ref)	1.00 (ref)	
Minor	1032	81	10.47	1.38 (1.08–1.77) *	1.48 (1.16–1.90) **	0.002
Major	90	11	20.88	3.00 (1.64–5.48) ***	1.87 (1.02–3.44) *	0.043
Vascular dementia						
Comparison	4488	95	2.40	1.00 (ref)	1.00 (ref)	
Minor	1032	23	2.91	1.30 (0.82–2.06)	1.36 (0.86–2.17)	0.188
Major	90	1	1.80	0.88 (0.12–6.31)	0.57 (0.08–4.11)	0.577

HR, hazard ratio; CI, confidence interval. * *p* < 0.05, ** *p* < 0.010, and *** *p* < 0.001.

## Data Availability

The data used in this study are owned by the Korean National Health Insurance Service. Data can be used with permission from the Korean National Health Insurance Service.

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
