# Peer review of "Risk of Dementia in Patients Who Underwent Surgery under Neuraxial Anesthesia: A Nationwide Cohort Study"

_jpm, 2021, doi:10.3390/jpm11121386_

Round 1

Reviewer 1 Report

  1. Methods: Please include surgery type  (small or major) of those patients. 
  2. Results: Figure 4 in legend: A and B were for subsequent figures in the next page? If so, please label clearly. 
  3. Discuss: Local anaesthetic toxicity is very likely irrelevant rather than surgery induced inflammation playing an important role in such setting. SO please cite those work (https://doi.org/10.1097/00000542-200703000-00007; doi: 10.1097/CCM.0b013e3181f17bcb; DOI: 10.1002/ana.22082; DOI: 10.1073/pnas.1014557107; doi: 10.1016/j.ebiom.2018.10.021; doi: 10.1097/SLA.0000000000005041; DOI: 10.1097/ccm.0b013e3181f17bcb; https://doi.org/10.1016/j.aat.2015.07.002) to discuss surgery/surgical trauma, systemic inflammation and neuroinflammation  vs neurological consequences and other mechanisms including metabolic changes and cite this (https://doi.org/10.1093/ageing/afz132) and other relevant references.
  4. Regarding post-operative management (treatment or prevention in older patient during perioprative period, those cite those (DOI: 10.1016/j.bja.2020.06.063; DOI: 10.1097/SLA.0000000000005287) to discuss.

Author Response

  1. Methods: Please include surgery type (small or major) of those patients.

Answer: Thank you for your good comment. We performed analysis of data according to surgery type (minor and major) as subgroup analysis. We added this contents in Methods and Results

  1. Results: Figure 4 in legend: A and B were for subsequent figures in the next page? If so, please label clearly.

Answer: Thank you for your good comment.  We labeled the figure 4.

  1. Discuss: Local anaesthetic toxicity is very likely irrelevant rather than surgery induced inflammation playing an important role in such setting. SO please cite those work (https://doi.org/10.1097/00000542-200703000-00007; doi: 10.1097/CCM.0b013e3181f17bcb; DOI: 10.1002/ana.22082; DOI: 10.1073/pnas.1014557107; doi: 10.1016/j.ebiom.2018.10.021; doi: 10.1097/SLA.0000000000005041; DOI: 10.1097/ccm.0b013e3181f17bcb; https://doi.org/10.1016/j.aat.2015.07.002) to discuss surgery/surgical trauma, systemic inflammation and neuroinflammation vs neurological consequences and other mechanisms including metabolic changes and cite this (https://doi.org/10.1093/ageing/afz132) and other relevant references.

Answer: Thank you for your good comment. We added contents including surgery/surgical trauma, systemic inflammation and neuroinflammation, neurological consequences and other mechanisms including metabolic changes.

Postoperative events can be a risk factor for dementia. In animals, postoperative cognitive dysfunction is associated with postoperative cytokine-induced inflammation in the hippocampus. [20, 21] Various proinflammatory cytokines, such as tumor necrosis factor alpha, maintain a state of chronic neuroinflammation, resulting in postoperative cognitive impairment and postoperative delirium. [22,23] Surgery may induce astrogliosis, β-amyloid accumulation and τ phosphorylation in the elderly, which may be associated with cognitive decline seen in postoperative cognitive dysfunction. [24] The decreased functional connectivity of the executive control network and its anticorrelation with the default mode network may contribute to executive function deficits following surgery. [25] The brain drives the surgical stress response by initiating changes in neuroendocrine balance, but changes in homeostasis can lead to postoperative cognitive impairment (POCD). [26] Changes in the metabolism before and after surgery can be related to postoperative delirium. [27]

  1. Regarding post-operative management (treatment or prevention in older patient during perioperative period, those cite those (DOI: 10.1016/j.bja.2020.06.063; DOI: 10.1097/SLA.0000000000005287) to discuss.

Answer: Thank you for your good comment. We added contents including perioperative management.

Perioperative management may be associated with the risk of dementia. Preoperative fasting, body temperature control, and blood pressure management are associated with the risk of postoperative cognitive disorder. [28] Patients with high risk of postoperative neurological complications are required frailty screening with preoperative cognitive screening for best perioperative neurological results. [29]

Reviewer 2 Report

Kwon et al. investigated the association between neuraxial anesthesia and the risk of dementia. Some study concerns should be mentioned.

  1. The rationale for linking neuraxial anesthesia and cognitive decline or dementia should be described more clearly. For general anesthesia, it is possible because of the intravenous anesthesia process. However, the author should demonstrate some evidence or literature to clarify the thinking process.

  1. The research selected non-anesthesia as the comparison group. Are the non-anesthesia subjects who underwent surgery? If not. The exposure factors between groups could be neuraxial anesthesia plus surgery versus non-anesthesia plus non-surgery. Therefore, local anesthesia as the active comparison group could be considered.

  1. The exclusion criteria included subjects who underwent other anesthesia besides neuraxial anesthesia from 2002 to 2013 and subjects with a history of brain and heart surgery from 2002 to 2013. Some issues needed to be discussed here. If we exclude some conditions after the index date, the study population will be more homogeneous. However, subjects were excluded after the index date is strange. Besides, suppose we exclude subjects with a history of brain and heart surgery after the index date. In that case, subjects with traumatic brain injury or stroke also need to be considered because of the potential factors related to dementia diagnosis in the future.

  1. How to select the variable in propensity score match? Why did the researcher perform multivariable-adjusted regression models in a propensity score matching cohort?

  1. How to validate the diagnosis of dementia, Alzheimer’s disease, and vascular dementia based on the ICD code? Besides, the national health insurance database underestimated cognitive dysfunction or dementia diagnosis, so hospital-based or registry-based datasets that include clinical cognitive function tests should be analyzed to confirm this finding.

Author Response

  1. The rationale for linking neuraxial anesthesia and cognitive decline or dementia should be described more clearly. For general anesthesia, it is possible because of the intravenous anesthesia process. However, the author should demonstrate some evidence or literature to clarify the thinking process.

Answer: Thank you for your good comment. We added contents including rationale for linking neuraxial anesthesia and cognitive decline or dementia.

Postoperative events can be a risk factor for dementia. In animals, postoperative cognitive dysfunction is associated with postoperative cytokine-induced inflammation in the hippocampus. [20, 21] Various proinflammatory cytokines, such as tumor necrosis factor alpha, maintain a state of chronic neuroinflammation, resulting in postoperative cognitive impairment and postoperative delirium. [22,23] Surgery may induce astrogliosis, β-amyloid accumulation and τ phosphorylation in the elderly, which may be associated with cognitive decline seen in postoperative cognitive dysfunction. [24] The decreased functional connectivity of the executive control network and its anticorrelation with the default mode network may contribute to executive function deficits following surgery. [25] The brain drives the surgical stress response by initiating changes in neuroendocrine balance, but changes in homeostasis can lead to postoperative cognitive impairment (POCD). [26] Changes in the metabolism before and after surgery can be related to postoperative delirium. [27] Perioperative management may be associated with the risk of dementia. Preoperative fasting, body temperature control, and blood pressure management are associated with the risk of postoperative cognitive disorder. [28] Patients with high risk of postoperative neurological complications are required frailty screening with preoperative cognitive screening for best perioperative neurological results. [29]

Reduction of cerebrospinal fluid pressure due to cerebrospinal fluid loss in the epidural space after spinal anesthesia is not uncommon. [30] In older people, there may be a persistent and significant decrease in cerebrospinal fluid pressure. [31] Patients with low cerebrospinal fluid flow have significantly reduced memory, visual construction, and verbal fluency. Alterations in cerebrospinal fluid flow may contribute to some of the cognitive deficits observed in Alzheimer's disease patients. [32] Injection of the drug into the lumbar epidural space compresses the dural sac, changes the compliance of the spinal subarachnoid space, and moves cerebrospinal fluid upwards towards the skull. Depending on the amount of the drug, intracranial pressure may increase. [33] The amount can be increased with prolonged anesthesia or with a patient-controlled analgesia catheter administered after nerve axis anesthesia. Increased intracranial pressure may be associated with cognitive impairment. [34] Spinal anesthesia triggers cerebral vasodilation [35] and may raise intracranial pressure. [36] In addition, the ability to control cerebral blood flow decreases in the elderly. [37]

  1. The research selected non-anesthesia as the comparison group. Are the non-anesthesia subjects who underwent surgery? If not. The exposure factors between groups could be neuraxial anesthesia plus surgery versus non-anesthesia plus non-surgery. Therefore, local anesthesia as the active comparison group could be considered.

Answer: Thank you for your excellent advice.  The comparison group is non-anesthesia regardless of surgery. We classified the two groups by code of anesthesia. The comparison may include surgery under local anesthesia. However, local anesthesia is usually performed in simple procedures including skin surgery (eg, skin biopsy, scar revision, small skin grafts), open wound repair, abscess drainage, foreign body removal from the skin, vascular access and dental procedures. [1,2,3] Most bleeding can be controlled with simple pressure on the wound [4], Infection, though relatively uncommon. Dehiscence (separation of wound edges) is infrequent.[5] Much postoperative care is not needed because complications are rare [5,6,7], and active pain control is usually not required because the surgical wound is small. Considering the various hemodynamic changes during nerve axis anesthesia, the possibility of complications and cost after neuraxial anesthesia, there is no benefit of neuraxial anesthesia for simple surgery. Therefore, it is unlikely that surgery of a scale that would require neuraxial anesthesia under local anesthesia.

It is difficult to consider local anesthesia as the active comparison group. There was no separate code for local anesthesia in KNHIS-NSC. Additionally, because local anesthetics can be used in treating cardiac arrhythmia and pain control, except for local anesthesia, it is difficult to classify with local anesthetics.

The contents of this have been added to the limitation.

Second, it was difficult to evaluate the effect of various procedures, including surgeries and local anesthesia, on our findings. We evaluated the effects of anesthesia and surgery on dementia as a series of procedures. However, surgery may affect immediate survival and long-term outcomes. [48] There are many types of surgery, and the corresponding intra- and post-operative effects may be different. The comparison group was non-anesthesia regardless of surgery. We classified the two groups by code of anesthesia, but there was no separate code for local anesthesia in KNHIS-NSC. The comparison group may include simple surgery under local anesthesia. Local anesthesia is usually administered for simple procedures, including skin surgery, open wound repair, abscess drainage, foreign body removal from the skin, vascular access, and dental procedures. [49-51] The direct effect on the brain by local anesthesia for simple surgery may be limited. In addition, intensive postoperative care is not needed because complications of simple surgery are rare, and active pain control is usually not required because the surgical wound is small. Although we used KNHIS–NSC dataset in the present study, it has been established for medical service claims and reimbursement, not for research. This dataset lacked sufficient data on procedures, including surgeries and local anesthesia. Additional studies are needed to evaluate the effects of each surgery and local anesthesia.

Reference

  • McGee, DL. Local and topical anesthesia. In: Clinical Procedures In Emergency Medicine, 5th edition, Roberts, JR, Hedges, JR (Eds), Saunders Elsevier, Philadelphia 2010. p.481.
  • Hruza, GJ. Anesthesia. In: Dermatology, 2nd, Bolognia, JL, Jorizzo, JL, Rapini, RP (Eds), Mosby Elsevier, Spain 2008. Vol 2, p.2173.
  • Bahl R. Local anesthesia in dentistry. Anesth Prog. 2004;51(4):138-142.
  • Skin biopsy: a field of interaction between clinician and pathologist.
  • https://www.uptodate.com/contents/skin-biopsy-techniques?search=skin%20biopsy&source=search_result&selectedTitle=1~150&usage_type=default&display_rank=1, access on 2021.12.10
  • Butler K. Incision and drainage. In: Clinical Procedures in Emergency Medicine, 5th ed, Roberts JR, Hedges JR (Eds), Saunder Elsevier, Philadelphia 2010. p.657
  • Daly L, Durani Y. Incision and drainage of a cutaneous abscess. In: Textbook of Pediatric Emergency Procedures, 2nd ed, King C, Henretig FM (Eds), Lippincott, Williams and Wilkins, Philadelphia 2008. p.1079.

  1. The exclusion criteria included subjects who underwent other anesthesia besides neuraxial anesthesia from 2002 to 2013 and subjects with a history of brain and heart surgery from 2002 to 2013. Some issues needed to be discussed here. If we exclude some conditions after the index date, the study population will be more homogeneous. However, subjects were excluded after the index date is strange. Besides, suppose we exclude subjects with a history of brain and heart surgery after the index date. In that case, subjects with traumatic brain injury or stroke also need to be considered because of the potential factors related to dementia diagnosis in the future.

Answer: Thank you for your advice. It is a very important matter.  We excluded heart and brain surgeries that could directly affect the brain. As you said, though postoperative traumatic brain injury or stroke are important, we could not consider it enough. Despite the importance of them, we only included them in the Charlson morbidity index. We described this content in the limitation part, and we will consider the effect of postoperative traumatic brain injury or stroke in further studies.

Third, our results are likely to be further biased due to confounding by indication or postoperative events, that is, patients who received neuraxial anesthesia may have done so because of unmeasured differences between groups, such as frailty, patient or clinician preference, or traumatic brain injury and stroke during the follow-up period. We matched the Charlson comorbidity index between the two groups to reduce the confounding as much as possible.

  1. How to select the variable in propensity score match? Why did the researcher perform multivariable-adjusted regression models in a propensity score matching cohort?

Answer:  Thank you for the good question. Among the variables found in KNHIS-NSC, we selected socio-demographic factors associated with dementia and comorbid variables.

If data with similar tendencies are extracted from the two groups using propensity score matching and logistic regression analysis is performed separately before and after matching, as a result, data after matching are more suitable for logistic regression analysis than data before matching. It can be confirmed by the -2Log Likelihood and Hosmer-Lomeshow method, which are conformity tests.[1]

Reference

  • Kim, So Youn, and Jong Il Baek. "On Logistic Regression Analysis Using Propensity Score Matching." Journal of Applied Reliability4 (2016): 323-330.

  1. How to validate the diagnosis of dementia, Alzheimer’s disease, and vascular dementia based on the ICD code? Besides, the national health insurance database underestimated cognitive dysfunction or dementia diagnosis, so hospital-based or registry-based datasets that include clinical cognitive function tests should be analyzed to confirm this finding.

Answer: Thank you for your good comment. This is a very important matter. We cannot know how to validate the diagnosis of dementia, Alzheimer’s disease, and vascular dementia with the ICD code. So, it may underestimate cognitive dysfunction or dementia diagnosis. As you said, the method to use hospital-based or registry-based datasets that include clinical cognitive function tests is a very excellent idea. We described this content in the limitation part. Because KNHIS-NSC data was anonymous, it is difficult to link hospital-based or registry-based datasets. Also, to link KNHIS-NSC data to hospital data or other registry data, additional permission is needed in the Republic of Korea.

Fourth, we could not validate the diagnosis of dementia, Alzheimer’s disease, and vascular dementia with the ICD code. Thus, we may have underestimated cognitive dysfunction or dementia diagnosis. Further studies are needed to analyze hospital-based or registry-based datasets that include clinical cognitive function tests to confirm our findings.

Round 2

Reviewer 2 Report

All comments had been replied to appropriately, and the researchers added these comments to the study limitation. I have no further suggestions.